# Change Detection in Synthetic Aperture Radar Images Based on a Generalized Gamma Deep Belief Networks

**DOI:** 10.3390/s21248290

**Published:** 2021-12-11

**Authors:** Meng Jia, Zhiqiang Zhao

**Affiliations:** Shaanxi Key Laboratory for Network Computing and Security Technology, School of Computer Science and Engineering, Xi’an University of Technology, NO. 5 South Jinhua Road, Xi’an 710048, China; zhaozq@xaut.edu.cn

**Keywords:** change detection, synthetic aperture radar (SAR), generalized Gamma deep belief network

## Abstract

Change detection from synthetic aperture radar (SAR) images is of great significance for natural environmental protection and human societal activity, which can be regarded as the process of assigning a class label (changed or unchanged) to each of the image pixels. This paper presents a novel classification technique to address the SAR change-detection task that employs a generalized Gamma deep belief network (gΓ-DBN) to learn features from difference images. We aim to develop a robust change detection method that can adapt to different types of scenarios for bitemporal co-registered Yellow River SAR image data set. This data set characterized by different looks, which means that the two images are affected by different levels of speckle. Widely used probability distributions offer limited accuracy for describing the opposite class pixels of difference images, making change detection entail greater difficulties. To address the issue, first, a gΓ-DBN can be constructed to extract the hierarchical features from raw data and fit the distribution of the difference images by means of a generalized Gamma distribution. Next, we propose learning the stacked spatial and temporal information extracted from various difference images by the gΓ-DBN. Consequently, a joint high-level representation can be effectively learned for the final change map. The visual and quantitative analysis results obtained on the Yellow River SAR image data set demonstrate the effectiveness and robustness of the proposed method.

## 1. Introduction

Change detection can be regarded as a classification procedure that classifies pixels into changed and unchanged classes. With the use of various feature learning and classification technologies, change detection can be used to acquire land cover change information from two images taken in the same area at two different times. This interesting task features a wide range of applications related to environmental monitoring [1,2], urban studies [3], forest monitoring and damage assessment [4,5], risk analysis, etc. SAR sensors are active microwave sensors and have been widely used for change detection tasks, because SAR images can be acquired under inclement weather conditions at any time. Thus, SAR image change-detection techniques are applicable in a wide range of fields. In this paper, we focus on the change detection in the Yellow River Estuary area related to changes in water and farmland by analyzing two synthetic aperture radar (SAR) images captured over the study area.

Monitoring changes in the Yellow River Estuary area of China is of great significance for human societal activity and natural resource protection. A large amount of sediment is deposited in the Yellow River channel and estuary area every year, which has changed the topography of the rivers and estuaries. In order to support navigation and production safety, changing topographical information is of significant value. In this paper, we focus on change detection in the Yellow River Estuary area, which is defined as identifying significantly changed areas in farmland, coastline and river by analyzing two SAR images captured over the same geographical area. The Yellow River Estuary data set, which comprises four-look data and a single-look data, is characterized by different looks, which means that the two images are affected by different levels of speckle. The huge difference in speckle noise level between the two images complicates the change detection process [6].

There has been a long-term effort to detect changes in SAR images in an unsupervised manner. Thresholding and clustering are two widely used classical algorithms for detecting changes in an unsupervised manner. However, the existence of speckle noise makes it difficult to separate opposite classes. Therefore, by incorporating spatial information, these two classical change-detection techniques can greatly improve the final detecting result. Jia et al. proposed to embed a priori knowledge into the expectation-maximization (EM) iteration process, which specifies the spatial characteristics of the pixel classes through Dempster–Shafer evidence theory [7]. Celik proposed a simple and effective satellite image change-detection algorithm that applies k-means clustering to principal component analysis (PCA) feature vectors constructed from nonoverlapping blocks of the absolute-value of difference in intensity of two images [8]. Local neighborhood information has also been incorporated into the multiple kernel k-means clustering objective function to resist speckle for SAR change detection [9]. Although traditional methods lead to great successes, their performance is limited by the artificially designed features extracted from SAR images.

Due to their capacity to learn multilevel feature representations, deep neural networks have received widespread attention in recent years. Classical deep neural works, such as convolutional neural networks (CNNs), deep belief networks (DBNs) and deep autoencoder networks (DAENs), have been constructed to address the SAR change detection problem. Gao et al. proposed introducing the dual-tree complex wavelet transform into CNNs for SAR change detection to effectively reduce the effect of speckle noise [10]. Focusing on the training dataset diversity, Samadi et al. proposed training the DBN using the input images and their morphological features [11]. By utilizing the pseudo labels obtained from a clustering algorithm, Gong et al. proposed constructing deep architectures using two steps: unsupervised feature learning and supervised fine-tuning [12]. In addition, considering that superpixels can tightly adhere to real change image boundaries, a stacked contractive autoencoder (sCAE) was presented to extract the temporal SAR image change feature [13]. Change detection using deep neural networks in SAR images is a complicated process, and it can be affected by many factors. The challenges for change detection are summarized as follows.

**Speckle noise**: speckling increases the overlap between opposite-class pixels in the histogram of the difference image, making it difficult to separate opposite classes. Existing change detection methods cannot detect all the ground object changes affected by speckle noise.

**Fuzzy edge of changed regions**: in detecting the change information from SAR images, changed regions may include a variety of ground object change information at the same time, whose characteristics are quite different. Due to the lack of prior knowledge, there is a competitive relationship between changed regions and background regions, resulting in a fuzzy edge in the changed region, which is difficult to determine. 

**Limited data set**: a large number of weights needs to be adjusted in the training process of deep neural network models, which makes it easy to cause the loss function to fall into the local minimum due to improper weights, resulting in poor change detection. To obtain ideal weights, enough training samples need to be fed into the deep neural network for model training. However, collecting labeled data is a time-consuming, laborious and even impractical task.

Due to the capacity of the DBN to learn the statistical characterization of SAR images, the dependencies among each unit of the observed variables are learned by the DBN to model the generative procedure of SAR images. Specifically, the DBN can be treated as a multi-layer generative model composed of restricted Boltzmann machines (RBM) [14] as its modules. The hierarchical structure of SAR images is built by DBNs with constraints $l_{1}$, $l_{2}$ and $l_{1/2}$ [15] for SAR image target recognition. Furthermore, a Wishart-DBN [16] was proposed for SAR image classification by employing prior knowledge of SAR images. The accuracy of many SAR image change detection methods relies on the accuracy of the given statistical model at expressing the changed information. This is because speckle noise existing in SAR images leads to a high level of uncertainty between changed and unchanged regions. Several widely used probability distributions for SAR image modeling can be viewed as special cases of the generalized Gamma distribution (gΓD), such as Rayleigh, exponential, Weibull and Gamma distributions. Therefore, the gΓD is considered to offer strong descriptive ability as the statistical model of difference images.

The two images of the Yellow River SAR image data set are single-look image and four-look image, respectively. This means that the influence of speckle noise on one image is much greater than on the other. The huge difference in the speckle noise level complicates the processing of change detection, since this increases the uncertainty between changed and unchanged pixels in the histogram of the difference images. In this paper, a gΓ-DBN is investigated for detecting changes in the Yellow River Estuary SAR image data set. Firstly, after studying the characteristics of various difference images, a gΓ-DBN was constructed to extract hierarchical features from raw data and fit the distribution of difference images. Next, the high-order statistical characteristics of changed and unchanged pixels in difference images were acquired by the constructed gΓ-DBN to provide a unique interpretation of changes and background from bitemporal SAR images. As a consequence, a final change map was generated based on the extracted discriminative information using gΓ-DBN.

## 2. Proposed Method

Let us consider two co-registered SAR images $X_{1}=\left\{x_{1,1},x_{1,1},\ldots ,x_{1,N}\right\}$ and $X_{2}=\left\{x_{2,1},x_{2,1},\ldots ,x_{2,N}\right\}$, which are composed of *N* pixels and acquired over the same area at two different times, $t_{1}$ and $t_{2}$, respectively. SAR image change detection can be regarded as a classification procedure, in which pixels in difference image $X=\left\{x_{1},x_{2},\ldots ,x_{N}\right\}\in R^{N}$ are classified as changed or unchanged. The general framework of the proposed method, which is composed of three steps, is presented in Figure 1.

Step 1 (Difference Image Generation): Three difference images are generated by the mean-ratio detector, the neighborhood-based ratio operator and the ratio operator, respectively.Step 2 (Training Sample Construction): Pixel vectors constructed by corresponding pixel patches from difference images are utilized as the to-be-selected training data. The PCA-kmeans algorithm is adopted to classify the pixel vectors into three clusters, of which those at a close distance from the cluster center are taken as training samples.Step 3 (Classification by gΓ-DBN): The training samples generated in Step 2 are fed into the gΓ-DBN for model training. After training, all the pixel vectors from the original difference images are fed into the learned gΓ-DBN for classification, before the final change map is generated.

### 2.1. Difference Images Generation

The generation of the difference image is usually the first step in the traditional change-detection process. The change information in radar backscatter can be obtained by comparing the intensity values between images taken on two dates. Potential change information can be reflected by all kinds of clues about real changes. A series of difference image generation techniques have been proposed. The mean-ratio detector preserves the mean value in the local region, which modifies the local texture and may be adverse to detecting changes [17]. The neighborhood-based ratio operator considers the scene heterogeneity in local areas and is not influenced by scenes with different kinds of changes [1]. The ratio operator can better adapt to the statistical characteristics of SAR data and is very resistant to calibration errors. However, it is quite sensitive to the presence of image speckles [18]. Difference image generation techniques feature their own strengths and yield effective results for change detection in SAR images; each technique inevitably leads to a loss of feature information reflecting the real change. To jointly learn features from various difference images, in this paper, various difference images are fully considered to capture high-level statistical feature representation. The training samples are constructed from the difference images. This approach is implemented to acquire better changed and unchanged information representations than that obtained from a single difference image.

In many traditional SAR image change-detection methods, the accuracy of the result depends on the given statistical model and whether the model can accurately express the changed information. This is because SAR images suffer from speckle noise, which increases the uncertainty between changed and unchanged pixels in the histogram of the difference images. Several probability distributions that are widely used for modeling the opposite class pixels of difference images offer limited accuracy for describing the change information between bitemporal SAR images. Gaussian, exponential, Rayleigh, Weibull or Gamma distributions may be suitable for some kinds of changed regions, but not for others. These can be viewed as special cases of a generalized Gamma distribution (gΓD) with different parameters. The generalized Gamma distribution (gΓD) is considered to feature strong descriptive ability as the statistical model of difference images. Therefore, in this paper, we propose to model the difference images using the gΓD to jointly learn statistical features from various difference images for difference scenarios. Probability density distributions of the opposite class pixels from the Farmland data set are plotted in Figure 2. After studying the characteristics of the opposite class pixels from various difference image pixels, the generalized Gamma distribution was found to accurately express the change information and background and was used as the statistical model for the difference images.

### 2.2. Training Sample Construction

Pixel vectors constructed by corresponding pixel patches from difference images were utilized as the to-be-selected training data. The PCA-*k*means algorithm was adopted to classify the pixel vectors into three clusters, of which those at a close distance from the cluster center were taken as the training samples. Let i∈{1, 2,…,n} be a set of integers indexing the N pixels. Image patches centered at the corresponding pixel $x_{i}$ are extracted from the three difference images. Let $P_{i}^{MR}$ represent an image patch centered at pixel $x_{i}$ in the mean-ratio detector $X_{MR}$. $P_{i}^{NR}$ and $P_{i}^{R}$ represent image patches centered at pixel $x_{i}$ in the neighbourhood-based ratio and radio operator, $X_{NR}$ and $X_{R}$, respectively. The size of each patch is ω×ω. All the difference image patches $X_{MR}$, $X_{NR}$ and $X_{R}$ corresponding to pixel $x_{i}$ are reshaped into pixel vectors and concatenated into a single pixel vector $P_{i}$. Thus, the size of $P_{i}$ is $3\times \omega^{2}$. After the obtaining pixel vectors of all the image pixels, a PCA-*k*means algorithm is applied to classify the pixel vectors into three groups: (1) the changed class, $\Omega_{c}$, among which pixels with a close distance to the cluster center. This means these pixels feature a high probability of being changed; (2) the unchanged class $\Omega_{u}$, among which pixels feature a high probability of being unchanged as well; and (3) the fuzzy class, $\Omega_{f}$. It is difficult to assign an accurate label to pixels belonging to $\Omega_{f}$, since the speckle noise increases the uncertainty between the opposite class pixels in the difference image. Therefore, pixels belonging to $\Omega_{c}$ and $\Omega_{u}$ are utilized to generate training set T. A smaller number of real samples with a high probability of constructing the training set could lead to overfitting of the model. By contrast, selecting a large number of samples results in some training samples featuring incorrect class labels, which would negatively affect model training. Furthermore, it should be noted that the changed and unchanged image samples featured an imbalance in their distribution. Therefore, we generated virtual samples based upon these real samples in training set T using the means proposed by Gao et al. to expand the training set T and ensure that the positive and negative samples were equally distributed [10]. Then samples in T were fed into the gΓ-DBN to train the constructed deep neural network model, which is described in the following subsection.

### 2.3. Classification by a Generalized Gamma Deep Belief Network

DBN can directly model the generative procedure of SAR images, and can effectively learn a statistical model from input data via nonlinear mapping. The SAR image change-detection task is widely considered a classification problem. Because the DBN can be used to learn the statistical dependencies among each unit of observed variables, a gΓD-based DBN is constructed by stacking the RBMs in a hierarchical manner to learn the discriminative information from difference images. Therefore, a DBN is suitable for capturing high-level discriminant features for land cover change detection. The aforementioned difference image generation techniques have been widely utilized in the traditional SAR image change detection techniques and have achieved great success. However, because of the existence of speckle noise, probability distributions offer limited accuracy in describing the change information between bitemporal SAR images. Thus, utilizing the high representation learning capacity of deep neural network models, a gΓB-RBM was used to learn the statistical dependencies between the visible variables and the hidden nodes for modeling the difference images [19].

The input pixel vector $P_{i\;}\in T$ was employed as n visible variables $v=\left(v_{1},v_{2},\ldots v_{n}\right)\in {\left[0,1\right]}^{n}$. In addition, m hidden nodes $h=\left(h_{1},h_{2},\ldots h_{m}\right)\in {\left\{0,1\right\}}^{m}$ were utilized to model the statistical relations between visible variables as the output of gΓB-RBM. The joint probability of gΓB-RBM can be expressed as:(1)$$p\left(v,h;\theta \right)=\frac{1}{Z\left(\theta \right)}\exp\left(-E\left(v,h;\theta \right)\right)$$
where E(v,h;θ) is the energy function, and θ represents the model parameters. The partition function Z(θ) can be calculated by $Z\left(\theta \right)=\sum_{v,h}\exp\left(-E\left(v,h;\theta \right)\right)$. The energy function of gΓB-RBM can be defined as:(2)$$E\left(v,h;\theta \right)=-\overset{m}{\underset{i=1}{\sum }}\overset{n}{\underset{j=1}{\sum }}W_{ij}h_{i}\ln v_{j}-\overset{n}{\underset{j=1}{\sum }}\left(v_{j}^{\beta }-b_{j}\ln v_{j}\right)-\overset{m}{\underset{i=1}{\sum }}c_{i}h_{i}$$
where θ={W,b, c}. And $W=\left(W_{ij}\right)$ is a weight matrix in which each element $W_{ij}$ is a real-valued weight associated with the edge between visible unit (input data $P_{j}$) and hidden unit $h_{i}$. $b=\left(b_{j}\right)$ and $c=\left(c_{i}\right)$ are biases associated with the visible and hidden nodes, for i∈{1, 2,…,m} and j∈{1, 2,…,n}, respectively. The value β is the power of gΓD [20]. Next, the conditional probability of gΓB-RBM is given by:(3)$$p\left(v_{j}=P_{j}|h;\theta \right)=\frac{p\left(v_{j}=P_{j},h;\theta \right)}{p\left(h;\theta \right)}\propto P_{j}{}^{b_{j}+\sum_{i=1}^{m}W_{ij}h_{i}}\exp\left(-P_{j}{}^{\beta }\right)$$
$$p\left(h_{i}=1|v;\theta \right)=\mathrm{sig}\left(\overset{n}{\underset{j=1}{\sum }}W_{ij}\ln v_{j}+b_{i}\right)$$
where sig(·) is the logistic function, defined as $\mathrm{sig}\left(x\right)={\left(1+\exp\left(-x\right)\right)}^{-1}$. Therefore, the probability that gΓB-RBM assigns between the visible and hidden nodes is determined by the input data $P_{j}$, weights W and biases b. Similar to [19,21], a batch-wise based gΓB-RBM training is implemented in Algorithm 1.

A discriminative gΓ-DBN that consists of an input layer, four hidden layers and one prediction layer is presented in Figure 1. The gΓ-DBN network architecture can be described as {I1, G2, B3, B4, B5, O6}. The value I1 is the input layer, where the input image pixel vectors are of the size n. The value G2 is a gΓB-RBM layer. The values B3, B4 and B5 are Binary-RBM layers. The value O6 is a softmax layer with two units to generate labels for change map. As a consequence, training samples in T are fed into the constructed gΓ-DBN for model training. Therefore, four major parts are concluded for the constructed gΓ-DBN training to deal with SAR image change detection task: gΓ-RBM learning, standard binary RBMs learning, adding a prediction layer and fine tuning, all of which tune weights of the gΓ-DBN with pseudo labeling information obtained from Step 2 (Training Samples Construction) via a backpropagation procedure. After training, all the pixel vectors from the original difference images are fed into the learned gΓ-DBN for classification, before the final change map is generated.
**Algorithm 1.** The gΓB-RBM update for a mini-batch of size Ns.**Input:** A gΓB-RBM with n visual nodes and m hidden nodes and training batch S.**Output:** The gradient approximation of model parameter:∆Wij, ∆bj and ∆ci, for i∈{1, 2,…,m} and j∈{1, 2,…,n}.  **Initialization:** ∆Wij=0, ∆bj=0 and ∆ci=0; **for** all v=(v1,v2,…vn)∈S **do**          v(0)←v;   **for** k=1 to K−1 **do**            ∀i∈{1, 2,…,m}, sample hi(k)~p(hi|v(k));            ∀j∈{1, 2,…,n}, sample vi(k+1)~p(vj|h(k));   **end for**   **for** 1≤i≤m and 1≤j≤n **do**    **Update** ∆Wij: ∆Wij←∆Wij+p(hi=1|v(0))·lnvj(0)−p(hi=1|v(K))·lnvj(K);    **Update** ∆bj: ∆bj←∆bj+lnvj(0)−lnvj(K);    **Update** ∆ci: ∆ci←∆ci+p(hi=1|v(0))−p(hi=1|v(K));   **end for****end for****return** ∆Wij, ∆bj and ∆ci.

## 3. Results

### 3.1. Experimental Setting

Quantitative and qualitative evaluations were utilized to compare the proposed method with related state-of-the-art methods on the Yellow River Estuary data set to demonstrate its effectiveness. This data set was acquired by the Radarsat-2 sensor in the C-band with polarization HH in “strip-map” mode over Dongying in Shandong Province, China, on 18 June 2008 and 19 June 2009, respectively. The spatial resolution was approximately 8 m × 8 m. More specifically, this data set is characterized by different looks. The image taken in 2008 is four-look data, but the one taken in 2009 is single-look data, which means that the two images are affected by different levels of speckle. The huge difference in speckle noise level between the two images complicates the change detection process. The original size of these two SAR images was 7666 × 7692 pixels. The details are difficult to illustrate; thus, we selected three typical areas located at different geographic sites with dissimilar types of changed regions. These three data sets were constructed by integrating prior information with photo interpretation, including Farmland, River and Coastline (detailed in reference [6]), as demonstrated in Figure 3. It is considered that the three regions can effectively reflect the changed characteristics of the Yellow River Estuary between two times. Figure 3a,b present a block of landlocked farmland, of which the changed regions are relatively large and regular. The available ground truth depicted in Figure 3c was created by integrating prior information with photo interpretation based on the input images. Figure 3d–f depict a section of an inland water area, which were selected because the changed regions are concentrated on the borderline of the Yellow River and comparatively difficult to accurately detect. The change in the coastline area is presented in Figure 3g–i, where the changed regions are on the surface of the sea, along the coastline. For this data set, the changed and unchanged pixels were distributed extremely unequally (1075 changed pixels and 124,925 unchanged pixels in the ground truth map).

To perform a broader comparison, four comparative methods were considered: (1) PCA-*k*means [8], which extracts eigenvectors with PCA and accomplishes change detection through *k*-means; (2) CWNN [10], which introduces dual-tree complex wavelet transform into CNNs for SAR change detection; (3) DBN [12], which obtains unsupervised feature learning and supervised deep belief network fine-tuning, then produces a final change map; and (4) JDBN, which jointly adopts three difference image samples as the input to train the DBN model. In the following implementations, image pixels from $\Omega_{u}$ and $\Omega_{c}$ considered as training samples were about thirty percent of the total image patch samples. Next, virtual samples were generated based upon these real samples in accordance with [10], to ensure that the positive and negative samples were equally distributed [10]. The power of gΓD β was set to 2, as recommended in the work of [19]. Note that, to make the comparison realistic, we applied the Lee filter on the bitemporal images to reduce the effect of speckle noise before generating the difference image.

As mentioned above, the gΓ-DBN network can be described as $\left\{I_{1},\;G_{2},\;B_{3},\;B_{4},\;B_{5},\;O_{6}\right\}$. The value $I_{1}$ is the input layer, where the input pixel vector $P_{i\;}$ is employed as n visible variables, which are determined by the size of the input image pixel vectors $3\times \omega^{2}$. The value $G_{2}$ is a gΓB-RBM layer with m hidden nodes, which is set to 170. The values $B_{3}$,$\;B_{4}$ and $B_{5}$ are Binary-RBM layers, in which the hidden nodes are fixed at 250, 200 and 100, respectively. The value $O_{6}$ is a softmax layer with two units to generate labels for change map. As a consequence, a 75-170-250-200-100-2 network is used. Every hidden layer is pretrained 50 passes through the entire training set, with the batch size $N_{s}$ being fixed at 100.

The performance of the-state-of-art change detection methods can be compared using visual and quantitative analyses after obtaining the final change detection map. Four quantitative evaluations were adopted in this study for performance evaluation: false positives ($N_{\mathrm{fp}}$), i.e., unchanged pixels that are identified as changed ones; false negatives ($N_{\mathrm{fn}}$), i.e., changed pixels that are categorized as unchanged ones; overall errors ($N_{\mathrm{oe}}$), i.e., the sum of $N_{\mathrm{fp}}$ and $N_{\mathrm{fn}}$; and kappa coefficient (κ) [22].

### 3.2. Reliability of the Training Sample Construction Method

A set of experiments were conducted to prove the reliability of PCA-kmeans algorithm by comparing with a supervised deep learning algorithm. The experiments were carried out on the three data sets (Farmland, Coastline and River), for each of which training and testing were respectively implemented on the same image in both methods. The samples used for training were selected by the rule described in “Step 2 (Training Samples Construction)”. The image pixels from $\Omega_{u}$ and $\Omega_{c}$ considered as training samples were about thirty percent of the total image patch samples. Next, virtual samples were generated based upon these real samples to ensure that the positive and negative samples were equally distributed. To ensure fairness, in the two methods, we used the same JDBN network topology and the same training set. However, it should be noted that the training set featured different labels in different methods. In the supervised method, the labels were given according to the ground truth, and in JDBN, the labels were given according to the pre-classification results. Although the testing set featured all the pixels of an image, we calculated the evaluation criteria by using the change detection results obtained from the JDBN.

Figure 4 presents the final maps of the three data sets. It can be seen that JDBN achieved similar results to the supervised method. After all, because the samples were from virtual samples in JDBN, there were some obvious false alarms or missed alarms. Furthermore, a quantitative comparison between the two methods on the three data sets is presented in Table 1. For the Farmland data set, the κ yielded by JDBN equaling to 0.8956 approached the value of 0.8931 obtained by the supervised method. For the River data set, the κ yielded by JDBN was 0.8019, a little lower than but close to the value of the supervised method. Furthermore, for the Coastline data set, the κ yielded by JDBN was 0.8932, similar to that of the supervised method. In conclusion, JDBN can exert similar effects to the supervised method on the change detection, which demonstrates that it is feasible to use the training sample construction method.

### 3.3. Performance of the Deep Learning Method

The goal of this section is to investigate the sensitivity of the proposed methods to the size of image patch ω, since this variable plays a critical role in network training. We demonstrate the effect of the image patch size ω on the SAR image change detection performance for gΓ-DBN using three real data sets, as described above. In these experiments, we evaluated the performance of the proposed method in varying the size of the image patch with 3×3, 5×5 and 7×7. Table 2 provides the change detection results under varying ω. Figure 5 depicts the final maps of the three sizes. It presents the worst performance when ω is set to 3. Because its change detection map features many white spots on the background.Making use of large image patches, noisy spots can be effectively suppressed due to the spatial information extracted by gΓ-DBN. With large image patches, the final maps produce many false alarms because of the loss of detailed information in the edge and texture regions. As demonstrated in the figure, the proposed method with ω being set to 5 exhibited the best performance in terms of $N_{\mathrm{oe}}$ and κ. Furthermore, ω was fixed at 5 in the following subsections.

### 3.4. Results and Analysis of the Real Data Sets

To evaluate the performance of the proposed gΓ-DBN in the Yellow River Estuary data set, we first conducted experiments on the Farmland data set. The quantitative results are presented in Table 3 and Figure 6a–f. Compared with that of other SAR image change-detection methods, i.e., the traditional PCA-*k*means method and three neural network-based methods, CWNN, DBN and JDBN, the performance of gΓ-DBN was good. As Table 3 demonstrates, CWNN and DBN feature a large $N_{\mathrm{oe}}$ and the $N_{\mathrm{oe}}$ for JDBN was better, up to 1112. This illustrates the performance of the joint feature learning strategy. The $N_{\mathrm{oe}}$ for gΓ-DBN was 1047, much better than that of other comparative methods. In addition, gΓ-DBN presented the best result in terms of κ (0.8956) because κ is a stationary term used to evaluate the agreement of the final change map. The value $N_{\mathrm{fp}}$ presents unchanged pixels that are identified as changed pixels. The final map features many white spots in the background, leading to a large value of $N_{\mathrm{fp}}$. Furthermore, $N_{\mathrm{fn}}$ denotes changed pixels that are categorized as unchanged pixels. The worst result for $N_{\mathrm{fn}}$ was caused by the loss of detail information in the edge and texture regions on the final change detection map. Additionally, the values of $N_{\mathrm{fn}}$ and $N_{\mathrm{fp}}$ jointly determined κ. These results indicate that gΓ-DBN outperformed DBN. Clearly, gΓ-DBN can provide more accurate statistical dependencies between the visible variables and the hidden nodes for difference images. The dependency exerted a considerable effect on the final change map. A direct comparison is shown in Figure 6b,f: fewer noise points were wrongly detected as changes by gΓ-DBN and PCA-*k*means.

The second data set used for performance comparison was the River data set, which reflects the change in a typical River area. Specifically, the bank of the River narrowed from 2008 to 2009, and an expanded pond was located in the bottom right corner. Table 4 lists the quantitative analysis results from the River data set. Figure 6g–l provide the change maps of all the comparison methods. Similar to the results from the Farmland data set, gΓ-DBN outperformed the other methods and achieved the best results in terms of $N_{\mathrm{oe}}$ and κ. Moreover, noise points appeared in the maps of the PCA-*k*means and CWNN, as highlighted in Figure 6h,i, leading to a large $N_{\mathrm{fp}}$. However, the edge regions on the change map of DBN were not well retained in Figure 6j, resulting in a large $N_{\mathrm{fn}}$.

Table 5 presents the quantitative evaluation results from the Coastline data set, and the visual results are provided in Figure 6m–r. For this data set, the changed area was relatively small. It can be observed that the $N_{\mathrm{oe}}$ results of these methods were relatively low. From Figure 6p,q,r, it is clear that the change maps lost detailed change information in the left corner, resulting in a large $N_{\mathrm{fn}}$. According to $N_{\mathrm{oe}}$ and κ, gΓ-DBN performed the best. As detailed in Figure 6n,o, more outliers were detected as changes by PCA-*k*means and CWNN because of the influence of speckle noise, leading to a large $N_{\mathrm{fp}}$.

## 4. Conclusions

This paper proposed a novel change-detection method for bitemporal SAR images of the Yellow River Estuary data set. The main contributions of this paper can be summarized in the following three aspects: firstly, in considering the gΓD’s strong ability to describe the statistical model of difference images, a gΓ-DBN was constructed to achieve more accurate statistical dependencies between the visible variables and the hidden nodes for the difference images. Secondly, the gΓ-DBN was trained in an unsupervised manner with the pseudo labeling technique due to the clustering algorithm and virtual samples to overcome the issue of limited training samples. Finally, the trained gΓ-DBN was utilized to jointly learn discriminative features from various difference images for the final change detection map. The experiments on the data sets with dissimilar types of changed regions demonstrate that the proposed gΓ-DBN method is superior to related methods, namely PCA-*k*means, CWNN, DBN and JDBN, at accurate change detection.

## Figures and Tables

**Figure 1 sensors-21-08290-f001:**
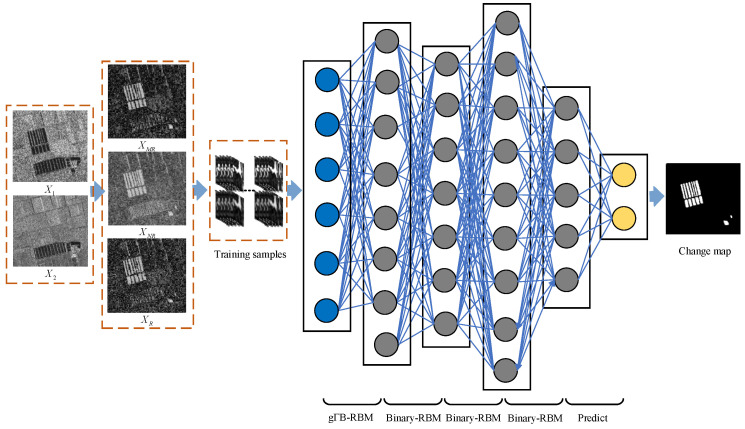
Illustration of the proposed framework.

**Figure 2 sensors-21-08290-f002:**
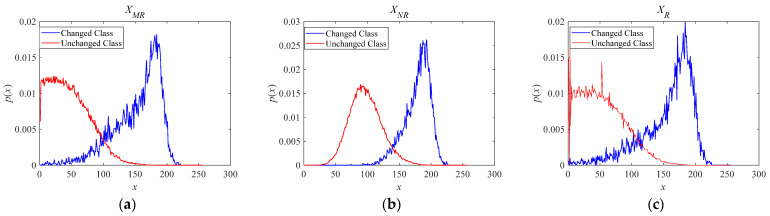
Histograms of opposite class pixels for various difference images obtained by (**a**) mean-ratio detector (**b**) neighborhood-based ratio operator and (**c**) ratio operator.

**Figure 3 sensors-21-08290-f003:**
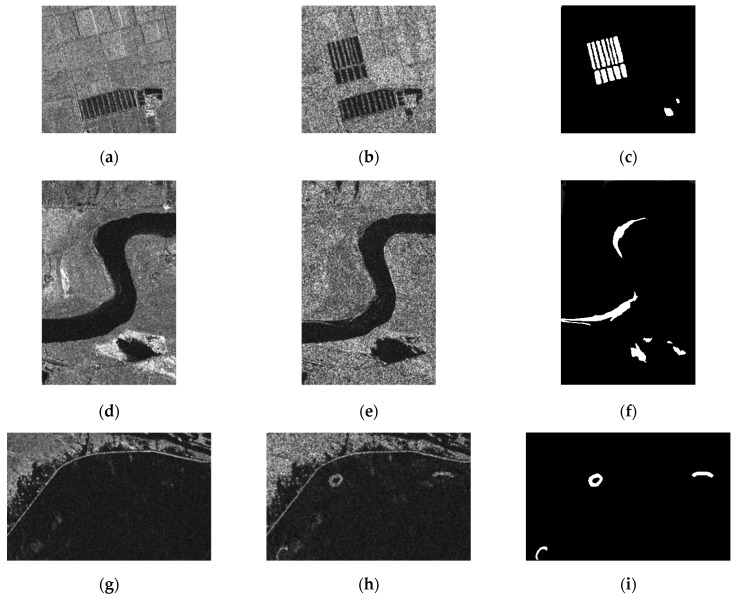
Presents the multi-temporal SAR image data sets used in experiments. Farmland data set with a size of 306 × 291 pixels. (**a**) Image acquired in June 2008. (**b**) Image acquired in June 2009. (**c**) Ground truth. River data set with a size of 291 × 444 pixels. (**d**) Image acquired in June 2008; (**e**) Image acquired in June2009; (**f**) Ground truth. Coastline data set with a size of 450 × 280 pixels. (**g**) Image acquired in June 2008; (**h**) Image acquired in June2009; (**i**) Ground truth.

**Figure 4 sensors-21-08290-f004:**
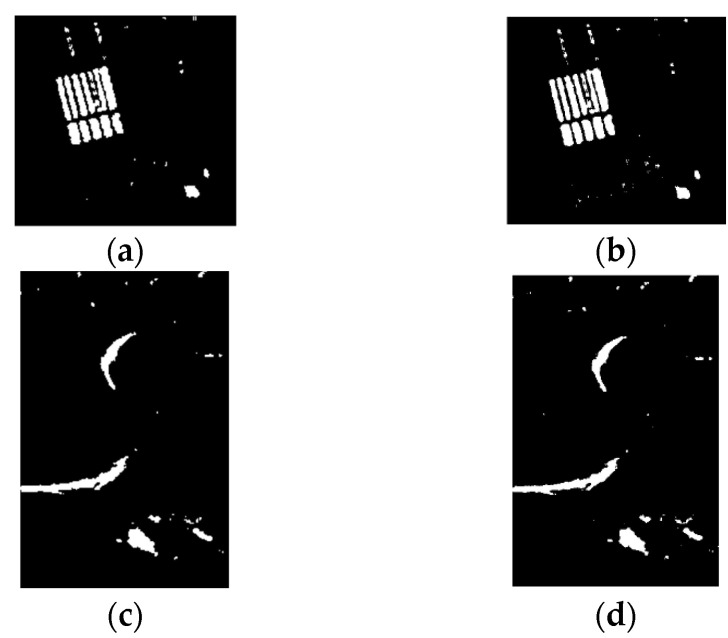
Change detection maps obtained by supervised and proposed methods on three SAR data sets. Farmland data set: (**a**) supervised method, (**b**) JDBN; River data set: (**c**) supervised method, (**d**) JDBN; Coastline data set: (**e**) supervised method, (**f**) JDBN.

**Figure 5 sensors-21-08290-f005:**
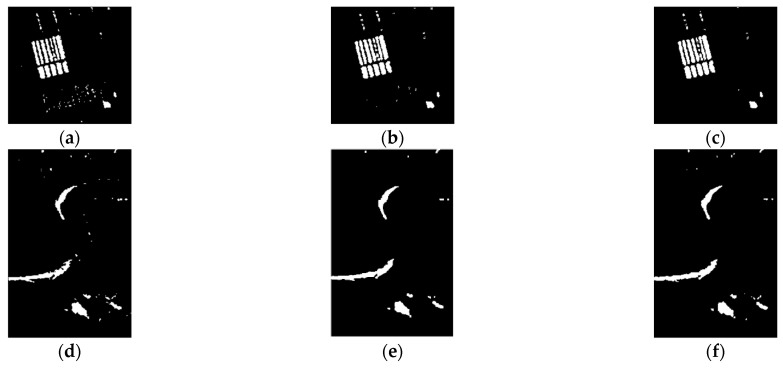
Change detection maps obtained by using different sizes of image patch in three SAR data sets. Farmland data set: (**a**) 3 × 3, (**b**) 5 × 5, (**c**) 7 × 7; River data set: (**d**) 3 × 3, (**e**) 5 × 5, (**f**) 7 × 7; Coastline data set: (**g**) 3 × 3, (**h**) 5 × 5, (**i**) 7 × 7.

**Figure 6 sensors-21-08290-f006:**
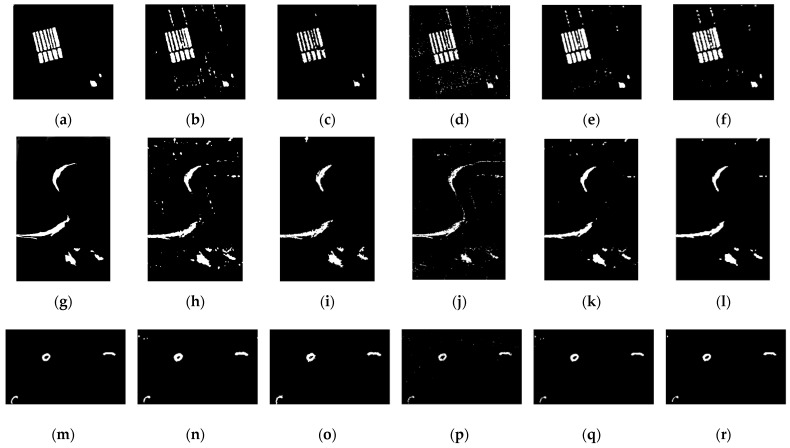
Depicts the change maps obtained by four comparative methods in three SAR image data sets. Farmland data set: (**a**) ground truth; (**b**) PCA-*k*means; (**c**) CWNN; (**d**) DBN; (**e**) JDBN; (**f**) gΓ-DBN. River data set: (**g**) ground truth; (**h**)PCA-*k*means; (**i**) CWNN; (**j**) DBN; (**k**) JDBN; (**l**) gΓ-DBN. Coastline data set: (**m**) ground truth; (**n**) PCA-*k*means; (**o**) CWNN; (**p**) DBN; (**q**) JDBN; (**r**) gΓ-DBN.

**Table 1 sensors-21-08290-t001:** Quantitative evaluation of the three data sets.

Data Sets	Methods	$N_{\mathrm{fn}}$	$N_{\mathrm{fp}}$	$N_{\mathrm{oe}}$	κ
Farmland data set	Supervised	469	604	1073	0.8931
	JDBN	456	656	1112	0.8898
River data set	Supervised	865	870	1795	0.7893
	JDBN	892	887	1779	0.7837
Coastline data set	Supervised	96	145	241	0.8894
	JDBN	99	145	244	0.8879

**Table 2 sensors-21-08290-t002:** Quantitative evaluation of image patch size ω in three data sets.

**Data Set**	3×3		5×5		7×7	
	$N_{\mathrm{oe}}$	κ	$N_{\mathrm{oe}}$	κ	$N_{\mathrm{oe}}$	κ
Farmland	1395	0.8597	1047	0.8956	944	0.9060
River	1843	0.7699	1595	0.8019	1664	0.7978
Coastline	238	0.8882	225	0.8932	255	0.8824

**Table 3 sensors-21-08290-t003:** Quantitative evaluation of the Farmland data set.

Methods	$N_{\mathrm{fn}}$	$N_{\mathrm{fp}}$	$N_{\mathrm{oe}}$	κ
PCA-kmeans	135	1440	1575	0.8366
CWNN	1570	57	1627	0.8104
DBN	1053	874	1927	0.8025
JDBN	456	656	1112	0.8898
gΓ-DBN	458	589	1047	0.8956

**Table 4 sensors-21-08290-t004:** Quantitative evaluation of the River data set.

Methods	$N_{\mathrm{fn}}$	$N_{\mathrm{fp}}$	$N_{\mathrm{oe}}$	κ
PCA-kmeans	581	2050	2631	0.7260
CWNN	722	1006	1728	0.7965
DBN	1298	842	2040	0.7258
JDBN	892	887	1779	0.7837
gΓ-DBN	892	703	1595	0.8019

**Table 5 sensors-21-08290-t005:** Quantitative evaluation of the Coastline data set.

Methods	$N_{\mathrm{fn}}$	$N_{\mathrm{fp}}$	$N_{\mathrm{oe}}$	κ
PCA-kmeans	29	445	474	0.8134
CWNN	22	375	397	0.8398
DBN	178	96	274	0.8664
JDBN	99	145	244	0.8879
gΓ-DBN	125	100	225	0.8932

## Data Availability

The authors would like to thank the Key Laboratory of Intelligent Perception and Image Understanding of the Ministry of Education of Xidian University for help in providing the test data sets.

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
