# Peer review of "Change Detection in Synthetic Aperture Radar Images Based on a Generalized Gamma Deep Belief Networks"

_sensors, 2021, doi:10.3390/s21248290_

Round 1
Reviewer 1 Report
The author uses the generalized Gamma Deep Belief Network (gΓ-DBN) to jointly learn features from the difference images to obtain the Yellow River Estuary change detection map, and improve the accuracy and robustness of the difference image classification. The revised opinions and existing questions are listed below:
- The paper is about multi-temporal spatiotemporal change detection, but only the two images are used. I don’t think the amount of data is sufficient.
- Clustering method is used to obtain the training samples. Could you please provide the reliability of this method?
- Is there a basis for the ratio of real data and expanded virtual data in the data set.
- Line 228: "three comparative methods are considered", but in the following sections, there are four methods which are compared.
- Heading 3.1 appears twice in the article.
Author Response
Dear Editor,
Please find our revised manuscript entitled "Detecting Changes of the Yellow River Estuary via SAR Images Based on a Generalized Gamma Deep Belief Network" (Manuscript No. sensors-1480321).
We appreciate very much the Editor and the reviewers for their time and effort devoted to the peer-review of this letter. Their constructive comments have guided us to further improve the quality of this letter. Below we provide our point-by-point responses to the comments. The sentences in italic are the reviewers' comments which are followed by our responses.
We are looking forward to your response.
Sincerely
Meng Jia1, ZhiQiang Zhao1
1 School of Computer Science and Engineering, Xi'an University of Technology, NO.5 South Jinhua Road, Xi'an, Shaanxi (710048), China.
Tel: +86 029 82312231
Fax: +86 029 82312231
Email: jiameng112@163.com

Reviewer 2 Report
Please see the attached file.

Author Response

(The authors gave the same response as above.)

Reviewer 3 Report
The authors presented a change detection method and tested their method with some specific SAR dataset. It would be more appropriate to put more emphasis on the novelties of the method rather than the dataset. For example, the title starts with “Detecting Changes of the Yellow River Estuary…”, the abstract starts with “Change detection of the Yellow River Estuary area of China”, the introduction also starts the same way “Monitoring the changes of the Yellow River Estuary area of China has great…”
I suggest putting more emphasis on the change detection algorithm, rather than the dataset itself.
In particular, in the 2nd paragraph of the introduction it is written that
“However, for the Yellow River Estuary area SAR image dataset, change detection entails considerable difficulties because SAR images obtained at two different times are affected by different speckle noise levels, which arise from the characteristics of different looks”
What makes this dataset that much different from the other SAR change detection tasks? As you have correctly mentioned in the fourth paragraph of the Introduction, there are so many state-of-the-art methods to tackle the SAR change detection problem using deep neural networks, so why this dataset is so specific?
Anyway, what is the main contribution of this work? Using the dataset or the proposed architecture? or the statistical distribution ? At line 76: “…in this paper, a gΓ-DBN is investigated for application in SAR image”: the authors should clearly highlight their contributions, for example using bullet points.
There is a very serious issue with the references in the paper. For example in page 2:
- Jia et al. proposed...
- ...characteristics of the pixel classes through Dempster-Shafer evidence theory7.
- dual-tree complex wavelet transform into CNNs for SAR change detection to effectively reduce the effect of speckle noise10.
- Focusing on the training dataset diversity, Samadi et al. proposed training the DBN using the input images and their morphological features11.
- …to extract the temporal SAR image change feature13.
- …
There are other mistakes as well and this has to be solved and this problem should not have happened if the authors had already double checked the final pdf before submission.
English language usage of the paper can also be improved. Try to improve the clarity of the text. For example at lines 68 and 69,
“The detection of changes in the Yellow River SAR image dataset must address the issue that different looks of the multitemporal SAR images lead to different speckle noise levels.” This sentence can be written in a simpler/shorter way (must address the issue can be omitted). Or at line 71, “Several widely used probability distributions have limited accuracy in describing the change information between multitemporal SAR images and can be viewed as special cases of a generalized Gamma distribution (gΓD) with different parameters.” : Try to decompose long sentences that do not have strong interrelation between their parts.
At line 93, what is meant by radio operator? I guess it’s supposed to be ratio.
Figure 2: x-axis should have the label (for example, if it is intensity between 0-255)
At like 228, “To achieve a broader comparison, three comparative methods are considered: PCA-kmeans8, CWNN10, DBN12 and JDBN that jointly adopts three difference image sample as the input to train the DBN model.: Apart from the reference problem, these names should have also been defined already; for example CW in CWNN stands for convolutional-wavelet neural networks and so on
The caption of figure 4 does not clearly describe the figure, (name the three dataset)
Line 331 : “5. Conclusions This section is not mandatory but can be added to the manuscript if the discussion is unusually long or complex.”. This has been left from the template!
Author Response

(The authors gave the same response as above.)

Round 2
Reviewer 1 Report
The revised paper has clear research objectives, clear demonstration ideas and complete conclusions. At the same time, thank you for answering my questions.
Reviewer 2 Report
The paper has been improved a lot with the authors' efforts.
Reviewer 3 Report
Congratulations on the successful revision and the previous comments have been addressed.